# Peer review of "Municipal Support of Diabetes Management in Daycare, Kindergarten and School: A Qualitative Study of Differences, Challenges and Potentials"

_healthcare, 2022, doi:10.3390/healthcare10081557_

Round 1
Reviewer 1 Report
The introduction is well written, and the topic is presented as exciting and relevant. It would be interesting to know, however, what other qualitative research has been conducted regarding the view of employees in municipal health care services and what the employee perspective, in general, may contribute to improving care.
Overall, the article lacks coherence. The research questions are not pointed out clearly. The study's aim was to highlight challenges and potentials regarding municipal support in relation to diabetes care for children in schools, kindergartens, and daycare in a Danish context. However, the presented results or conclusion do not refer to potentials found, nor does it become evident what the barriers (to what?) are. Are they the same as the challenges found?
In the methods section, it does not become clear which interview approach the authors followed. Also, I do miss a reference regarding the study's descriptive design. A more detailed table describing the participants and their occupations would be helpful. For example, the results section suggests that some are nurses, which has not been introduced in the participant's section.
It does not become evident why the interviews are video recorded. What additional value does the video recording have? The coding process and who was coding are not clearly described. Then several code groups are mentioned. (How) are they reflected in the later presentation of results? How many levels of categories (or codes) have been established? What is meant by code groups? What was the research question that guided the analysis? Was coding inductive or deductive or both? An example of how the code groups and subgroups were condensed into the central theme would be helpful.
The results are not reported clearly. There are four headlines which refer to the four main themes. However, there are no subthemes; overall, the section lacks structure. A summary at the end of each section that refers to a central theme would provide more clarity.
The discussion then points out that the study found four deficiencies in the support. But how do these relate to the research question and identified main themes? Also, I would find it helpful if each of these four deficiencies were discussed in a more structured way.
The citation style changes throughout the discussion, and a name seems to have slipped into the last paragraph. Since it says, more in-depth studies are needed to describe the mechanisms revealed in our research accurately: What should these studies explore? What information is missing? What are the limitations of the study?
Author Response
The introduction is well written, and the topic is presented as exciting and relevant. It would be interesting to know, however, what other qualitative research has been conducted regarding the view of employees in municipal health care services and what the employee perspective, in general, may contribute to improving care.We have now stated that no prior qualitative research has been made in this specific area.
Overall, the article lacks coherence. The research questions are not pointed out clearly. The study's aim was to highlight challenges and potentials regarding municipal support in relation to diabetes care for children in schools, kindergartens, and daycare in a Danish context. However, the presented results or conclusion do not refer to potentials found, nor does it become evident what the barriers (to what?) are. Are they the same as the challenges found? We are now using “objective” instead of “aim” as we consider the objective as the research questions of this study. Regarding the results on potentials, we have now made this clearer throughout the result section.
In the methods section, it does not become clear which interview approach the authors followed. Also, I do miss a reference regarding the study's descriptive design. A more detailed table describing the participants and their occupations would be helpful. For example, the results section suggests that some are nurses, which has not been introduced in the participant's section. We have already stated that the interview approach was semi-structured. We have now made a detailed table with descriptions of the participants and their occupations.
It does not become evident why the interviews are video recorded. What additional value does the video recording have? The coding process and who was coding are not clearly described. Then several code groups are mentioned. (How) are they reflected in the later presentation of results? How many levels of categories (or codes) have been established? What is meant by code groups? What was the research question that guided the analysis? Was coding inductive or deductive or both? An example of how the code groups and subgroups were condensed into the central theme would be helpful. We have now elaborated written that the video method was enhanced for sake of the interview setting and not for the analysis. We have added a table with all the subgroups.
The results are not reported clearly. There are four headlines which refer to the four main themes. However, there are no subthemes; overall, the section lacks structure. A summary at the end of each section that refers to a central theme would provide more clarity. We have now added a short summary at the end of each section.
The discussion then points out that the study found four deficiencies in the support. But how do these relate to the research question and identified main themes? Also, I would find it helpful if each of these four deficiencies were discussed in a more structured way. We have now related the four deficiencies more to the identified main themes and structured the discussion accordingly.
The citation style changes throughout the discussion, and a name seems to have slipped into the last paragraph. Since it says, more in-depth studies are needed to describe the mechanisms revealed in our research accurately: What should these studies explore? What information is missing? What are the limitations of the study? The citations have been changed and the name has been removed. We have elaborated on what futures studies could explore and what the limitations are of our study.

Reviewer 2 Report
- The description of the methods is not very clear, it would be better to insert a table and a copy of main questions of the interview.
- In all municipalities the age range of type 1 patients was the same?
- 80 semi-structured online interviews with 121 municipal employees from 74 (of 98) municipalities were conducted. In some cases more employers by the same municipalities partecipates: it's better to detail.
- Did you find any difference in critical points depending on the cure (injections vs pump)?
Author Response
- The description of the methods is not very clear, it would be better to insert a table and a copy of main questions of the interview. The interviews were only semi-structured and the main themes are stated in the article.
- In all municipalities the age range of type 1 patients was the same? We are not sure that we understand this question. There is no age-defined target group in the study
- 80 semi-structured online interviews with 121 municipal employees from 74 (of 98) municipalities were conducted. In some cases more employers by the same municipalities partecipates: it's better to detail. We have now added that more than one employee participated in some municipalities.
- Did you find any difference in critical points depending on the cure (injections vs pump)? This in an interesting question but this was not the focus of present study.

Round 2
Reviewer 2 Report
No more comments to the authors